# Analysis of Gas Turbine Compressor Performance after a Major Maintenance Operation Using an Autoencoder Architecture

**DOI:** 10.3390/s23031236

**Published:** 2023-01-21

**Authors:** Martí de Castro-Cros, Manel Velasco, Cecilio Angulo

**Affiliations:** Intelligent Data Science and Artificial Intelligence Research Centre (IDEAI), Automatic Control Department, Universitat Politècnica de Catalunya, Campus Nord, Carrer de Jordi Girona, 1, 3, 08034 Barcelona, Spain

**Keywords:** artificial intelligence, autoencoder, condition assessment, gas turbine, compressor

## Abstract

Machine learning algorithms and the increasing availability of data have radically changed the way how decisions are made in today’s Industry. A wide range of algorithms are being used to monitor industrial processes and predict process variables that are difficult to be measured. Maintenance operations are mandatory to tackle in all industrial equipment. It is well known that a huge amount of money is invested in operational and maintenance actions in industrial gas turbines (IGTs). In this paper, two variations of autoencoders were used to analyse the performance of an IGT after major maintenance. The data used to analyse IGT conditions were ambient factors, and measurements were performed using several sensors located along the compressor. The condition assessment of the industrial gas turbine compressor revealed significant changes in its operation point after major maintenance; thus, this indicates the need to update the internal operating models to suit the new operational mode as well as the effectiveness of autoencoder-based models in feature extraction. Even though the processing performance was not compromised, the results showed how this autoencoder approach can help to define an indicator of the compressor behaviour in long-term performance.

## 1. Introduction

Industrial gas turbines (IGTs) for power generation are a kind of internal combustion engine that converts the chemical energy of fuel into electrical power. They are mainly composed of three components: compressor, combustor, and power turbine [1]. The use of IGTs is widespread and range in size from small and mobile power plant units to high and static systems with large productions. Gas turbines can be financially justified in a short profitable service time; thus, they are more economically compatible with a decarbonising grid than other types of fossil fuel infrastructure [2]. However, in view of the uncertainties related to global warming, there is a requirement of monitoring and reduce toxic and noxious emissions [3].

IGTs are used to operate under harsh conditions. Several causes have been found that greatly damage the gas turbine system, indeed giving rise to malfunctioning events and deterioration [4]: fouling, corrosion, erosion, abrasion, and unexpected particles. Besides these main deterioration causes, several other fatigue factors could be considered as being harmful to the system performance due to changes in the standard operational regime, such as the number of starts and stops or modifications in the output power set point. Figure 1 shows a unit of the used gas turbines to perform the current analysis.

Maintenance is an important regular task to preserve the good performance of the machine and to avoid structural degradation that leads to system breakdowns. Major maintenance operations, however, restore IGTs not to their new conditions but to a new operational point, thus ensuring no malfunctioning events and reducing deterioration [5].

Operational changes due to both major maintenance operations and user decisions beyond the nominal operational regime are currently real concerns in design and process engineering. The current trend in effective maintenance strategies is beyond the restoration of the system to a profitable operational point. Maintenance operations must also consider the economical, environmental, and security implications from unexpected events in the system [6]. Hence, this work aims to introduce a tool in the form of a data-based machine learning scheme able to analyse gas turbine performance, especially after a major maintenance operation, assessing engineers about the current condition of the process. Most of the methods related to this put the focus on anomaly detection, but none of them considers the whole operation of a gas turbine using only compressor data. The increase in the available data and the development of intelligent tools for condition and fault assessment have made this possible. The huge volume of data obtained from IGTs provide knowledge and information about the condition of components, malfunctioning events, and warnings from which the state of the system can be determined [7].

The analysis of captured data is mainly based on machine learning algorithms. These are able to extract information from multidimensional time series to automatically learn insights and recognise hidden patterns such as system performance, to increasingly make better decisions, etc. These kinds of methods are currently applied to a wide range of fields for automated analytical model building [8].

In this study, an autoEncoder (AE) architecture based on artificial neural networks (ANNs) is proposed to identify significant hidden patterns to determine operational changes in gas turbine performance based on compressor data. This kind of ANN-based architecture is trained to copy its inputs to its outputs [9]. Its structure is composed of two main elements, usually two ANNs: an encoder and a decoder, as shown in Figure 2. The encoder network *G*, or simply the encoder, is defined as an encoding function z=G(x), where x is the model input and z is a set of latent variables. The decoder network *F*, i.e., the decoder, is defined to reconstruct the encoded signal, $x^{\prime }=F\left(z\right)$, where $x^{\prime }$ refers to the reconstructed input signals. The set of weights for both ANNs are simultaneously learned by minimising a loss function $\epsilon =d(x,x^{\prime })$ according to some distance metric [10].

One of the main uses of AE is nonlinear dimensionality reduction, which is mostly used for the visualisation of high-dimensional data. Furthermore, it is shown that this kind of ANN-based architecture is able to learn deep representation features of the data. Thereby, the AE creates reliable models of complex systems [11], which can be applied for several purposes. Many AE architecture models and their variants have been successfully applied to many different gas turbine components mainly for operational anomaly detection. Marzieh Farahani [12] proposed an autoencoder model to have feature selection by keeping vital features and learning time series encoded representation. Afterwards, the encoded features were used together with an LSTM model to detect the anomaly sensors related to a specific gas turbine. A re-optimised deep autoencoder (R-DAE) was developed by Fu et al. [13] in the field of anomaly detection. The proposed model was improved in comparison with the existing unsupervised anomaly detection methods by automatically removing abnormal samples from the original training set and by using several samples from multiple engines to avoid the over-fitting problem. In the same field, in [14], a stacked denoising autoencoder (SDAE) was proposed to accurately detect anomalies in a gas turbine combustor. An intelligent monitoring model based on a stacked sparse autoencoder (SSAE) was introduced by Han et al. [15] to identify the combustion stability as well as to optimise the operating conditions. The combustion stability was also tackled using visual information.

Moreover, autoencoders were also applied to detect significant changes in the system condition or to determine the expected life of a component. Xu et al. [16] used a moving window-based autoencoder (MASAE) to construct a health indicator for predicting the remaining useful life (RUL) of roller bearings. A combination of several autoencoders and sliding windows architecture is proposed in a study by Barrera et al. [17] that aims to build a solution for detecting when a gas turbine presents abnormal behaviour. The authors highlight that the innovation of the method lies in not requiring existing disruption data, which is not limited to any time window, and it provides crucial information in real time to monitor operation. The so-called 3D convolution selective autoencoder (3DCSAE) method was used in [18] to capture the transition from stable to unstable regimes in combustion systems. The model was trained using only completely stable data and completely unstable data, so they showed that the technique is able to properly generalise the condition by identifying the gradual transitions.

The studies mentioned above focus on either the condition of a specific component or the expected life of the gas turbine. A significant issue left to study is the effect of major maintenance on gas turbine operation and its implications. Therefore, the general objective of this study was to analyse the condition of gas turbines after major maintenance by processing only compressor data in an IGT using data analytics and machine learning tools. Furthermore, we aimed to determine the effectiveness of AE-based models in the field of IGT maintenance.

These tools were developed and customised using real data, particularly from the compressor. Data were collected from sensors at different locations in the IGT’s compressor. The results were obtained by modelling the IGT’s compressor for a short-term period with a uniform regular condition in the form of an autoencoder. Therefore, the autoencoder modelling the compressor in the short term, $AE_{M}$, was trained on regular fresh conditions M so that $\epsilon_{M}=d\left(x_{M},x_{M}^{\prime }\right)=d\left(x_{M},AE_{M}\left(x_{M}\right)\right)$ would be minimised. Then, this model of a regular fresh compressor encoded in $AE_{M}$ was used to analyse the long-term performance of the industrial gas turbine’s compressor, P, $x_{P}^{\prime }=AE_{M}\left(x_{P}\right)$. The main analysis of the condition involved the deviation between the original data and the one obtained from the fresh model, $\epsilon_{P}=d\left(x_{P},x_{P}^{\prime }\right)=d\left(x_{P},AE_{M}\left(x_{P}\right)\right)$. Hence, this deviation is not an error measure, as it was during the training phase on the fresh regular data, but a discrepancy measure. From this discrepancy measure, several important insights that affect the efficiency and drift of the engine could be concluded as well as the way how the compressor’s gas turbine is displayed in operation.

The rest of the paper is organised as follows: In Section 2, a description of a selective series of autoencoder models is presented, as well as how data are treated and analysed to create the needed datasets. Next, in Section 3, the results obtained using this methodology are presented. Several analytical approaches are considered. Finally, in Section 4, these results are discussed, and conclusions are drawn in Section 5.

## 2. Materials and Methods

Machine learning is a general computational algorithmic approach currently applied in a wide range of fields for automated analytical model building. In this study, an autoencoder-based architecture is used to assess and analyse the behaviour of an IGT [19]. Next, a short introduction is provided about the proposed autoencoder model.

### 2.1. Model Description

An autoencoder is an unsupervised learning architecture that is used in a wide range of fields. Its goal is to set the target values so that they are equal to the original input data by learning effective encoding of original data in the form of input vectors [20]. The structure of this kind of method is two-faced artificial neural networks called encoder and decoder, as it was shown in Figure 2. As their names state, the encoder aims to learn a codified version of input data, while the decoder rebuilds the original data from the encoded one.

The output layer of the encoder, indeed the input layer of the decoder, is a vector formed from the compression/transformation of input data. This is of interest since this latent space gathers all the relevant information of input data in reduced dimensionality. This layer is also named *code*.

Autoencoders can be structured in several different ways depending on the problem at hand. In this case, two approximations are employed: a fully connected (FC) system and a sparse autoencoder. FC models are the simplest since all the neurons in the ANN are connected without any kind of constraint. Contrarily, the main characteristic of sparse networks is that a sparsity constraint fixed by a sparsity parameter ρ is applied in all network levels. Sparsity constraint consists of reducing node activation.

There are several hyperparameters that must be set in this kind of architecture. Some of them are the same as in ANN models, such as the number of hidden layers, the number of nodes per layer, the activation function, and the loss function employed to measure the match between the input data and recovered data. An extra parameter has to be set, which is the code size in the latent space. In this article, one layer was used for both the encoder and decoder, the number of nodes was set to five and eight for the input and hidden layer, respectively, and the activation function was set to be the rectified linear unit (ReLU), and the loss function under consideration was the mean-squared error (MSE) function. The code size was evaluated in two different values to further analyse the model performance; indeed, it was set to both 2 and 3. Therefore, the final autoencoder structures were 5-8-3-8-5 and 5-8-2-8-5, which are graphically represented in Figure 3a and Figure 3b, respectively. A very simple architecture was intentionally defined because our industrial data owner partner was interested in further research about explainability using the information from the latent space. Although explainability was out of the scope of this research, it was verified that no better results would be obtained with larger structures.

Each square refers to a neuron of the AE architecture model and each part is defined as follows: the green part corresponds to the sampled sensors of the gas turbine’s compressor, further explained in Section 2.2, while the yellow part refers to the hidden layers of eight dimensions, and the red part is the latent space set to either 2D or 3D depending on the model.

### 2.2. Data Processing

The main parameters to consider in a gas turbine’s compressor are its pressure and temperature. Thereby, the model inputs $x=(x^{\left(1\right)},\ldots ,x^{\left(5\right)})\in X$ were sampled sensors of an industrial gas turbine’s compressor placed strategically to capture these features, where

Inlet pressure (IP), $P_{c,i}$;Inlet temperature (IT), $T_{c,i}$;Relative ambient humidity (AH), $H_{a}$;Pressure ratio (PR), $rP_{c}=P_{c,o}/P_{c,i}$;Outlet temperature (OT), $T_{c,o}$.

The values for the five selected features were preprocessed before the model training procedure. Firstly, data were only collected under the *full-load* working regime. A full load is an operation mode such that the gas turbine is working at its limit condition, where degradation and faults become apparent. Thereby, this leads to a clear understanding and analysis of how gas turbines are performing at their maximum capacity. There exists an operation mode called *grid-code regulation mode* where the machine regulates the inlet air flow by itself in order to avoid harmful events when it is working at full load. These events were not considered in the analysis performed in this study; therefore, they were also filtered by applying a threshold in the *variable guide vane* (VGV) aperture. The threshold was defined using the median of the VGV aperture in full-load conditions for each piece of equipment. A set of ten gas turbines were considered to train the gas turbine model, but only the one with the highest quantity of data under the previous constraints was used to obtain the results. With these analyses, we aimed to obtain a robust enough model that captures the behaviour of gas turbine operation.

Next, a filtering method was applied for data cleaning to erase the most evident outlier elements. The filtering method consisted of computing the median for each feature and adding or subtracting *k*-times the median absolute deviation (MAD) to define both upper and lower boundaries. It was carefully defined to avoid finding anomalies inside the data. The median is a statistical method of measurement that separates half of the probability distribution, and MAD is a measure of the variability of a univariate sample of quantitative data. They are closely related to the mean and standard deviation (STD) measures; however, MAD is considered a more robust estimator in presence of outliers than STD, and the median is also more robust than the mean. The *k*-times constant was experimentally determined by comparing several gas turbine sensor graphs. The parameter value that gave the best performance in filtering data, indeed erasing the most significant outliers without deleting relevant correlation in data, was 7 times greater or lower than the median corresponding to either the upper or lower boundary.

Finally, standard scaling was used to normalise the range of independent variables or features of data with mean zero and standard deviation since the ranges between the features were too large and diverse from each other, and this relative variation could lead to a significant modelling error.

### 2.3. Dataset

As mentioned before, data were gathered from real gas turbine plants through sensors located at strategic positions in an industrial gas turbine’s compressor. Furthermore, some extra information was provided regarding several maintenance operations performed on the gas turbine’s compressor and their extent, regardless of whether the action taken was a simple inspection or major maintenance in a location where some important pieces were replaced.

#### 2.3.1. Training Set

The training set corresponds to the first year of gas turbine data,
(1)$$M={\left\{x\left(t_{k}^{m}\right)\right\}}_{k=1}^{N_{m}}={\left\{x_{k}\right\}}_{k=n_{m}}^{n_{m}+N_{m}}={\left\{m_{k}\right\}}_{k=1}^{N_{m}}\subset D\subset X,$$
with inputs x∈X as training features. The aim when selecting this period of time for training the model was to capture the behaviour of the new machine at the beginning of its life. This model served as a behavioural baseline to be compared with the performance in the long term, and a proper analysis of the compressor’s current working mode was conducted.
(2)$$AE_{M}=\left(G_{M},H_{M}\right),$$
where M⊂X is the fresh training data.

The sampling time $T_{s}$ to generate the training dataset was 1 min. Therefore, note that $x_{k}=x\left(t_{k}\right)$ and $x_{k+1}=x\left(t_{k+1}\right)=x\left(t_{k}+T_{s}\right)$.

#### 2.3.2. Validation Set

In this study, the validation set was equal to the training set. The model was constructed under the assumption that the equipment would be working under new conditions for the first working year. Basically, it was assumed that the gas turbine was in operation at its best condition. The loss metric
(3)$$\epsilon_{M}\left(x\right)=d\left(x_{M},AE_{M}\left(x\right)\right)$$
refers to the reconstruction distance when the autoencoder $AE_{M}=\left(G_{M},H_{M}\right)$ is trained with the modelling data $M={\left\{m_{k}\right\}}_{k=1}^{N_{m}}$. It is expected that $\epsilon_{M}\left(m_{k}\right)\approx 0$, for $m_{k}\in M$, as data were reconstructed in similar conditions to those used for training the model.

#### 2.3.3. Testing Set

The test data were defined as
(4)$$T={\left\{x\left(t_{k}\right)\right\}}_{k=1}^{N}={\left\{x_{k}\right\}}_{k=1}^{N}={\left\{t_{k}\right\}}_{k=1}^{N}\subset D\subset X,$$
represent the whole dataset, i.e., one that includes the entire available data for the assessed gas turbine, which was used to test the model of the compressor in terms of its long-term performance. Hence, the testing set contained data for three moments: the new gas turbine (first year) with training/validation set; data for the older turbine (more than the first year) but before any major maintenance; and data after major maintenance. As it was previously mentioned, it was assumed that the reconstruction was almost perfect for the training/validation set. Now, it was also expected that reconstruction would slowly degrade and not as much for testing data after the first year but rather before major maintenance. Finally, our discrepancy measure would prove effective if it is able to show a discrepancy, that is, a high degradation in the reconstruction after major maintenance.

#### 2.3.4. Maintenance Set

Maintenance events can be classified as inspections (I) and replacements (R). I refers to a periodic revision performed in the compressor to ensure the minimum quality component conditions, whereas R corresponds to a more complex operation, during which some components are changed and, therefore, the performance of the IGT is altered. It was expected that our autoencoder trained on new first-year data should be able to identify changes in the operational point in data after major maintenance (post-maintenance data) by showing a certain degree of discrepancy between the original data and the reconstructed data. By contrast, no abrupt changes in the operational point were expected after inspections; that is, the discrepancy should remain almost stable after inspections. This can be noticed in Figure 4.

## 3. Results

Four different kinds of autoencoders were used to perform the condition analysis, which differed in their structure and latent dimensions. In terms of structure, both a fully connected and sparse structures were selected. The criterion that determined the latent space dimension was defined by the lowest possible values with maximum model performance based on experimental results. These led to both two- and three-dimensional latent spaces. A summary of the employed nomenclature for these autoencoders is shown in Table 1.

As explained in Section 2.3, three different kinds of datasets were used to develop and assess the gas turbine model. All the analyses of gas turbines used the same data. Firstly, the training set included the first-year data, from 3 January 2017 to 3 January 2018, which were used to train the models. Then, the testing set involved the remaining data, from 4 January 2018 to 1 January 2020, which made it possible to obtain the results from every model. Finally, the maintenance set included major and minor maintenance operations, and inspection events were also considered (they are marked with a green line) to determine how operations changed as a result of these events.

An experiment was carried out to analyse the IGT compressor’s condition and detect changes in the operational point using the root-mean-squared error (RMSE). The RMSE was used to measure the discrepancy distance (Equation (Equation 3)) between the autoencoder’s inputs and outputs. It is called *reconstruction discrepancy (RD)*,
(5)$$RD=\frac{1}{n}\sum_{i=1}^{N}\sqrt{{\left(x_{i}-x_{i}^{\prime }\right)}^{2}}$$
where *x*
ϵ
T and $x_{i}^{\prime }$ are the output of the corresponding AE model. For obtaining the results, the data-grouping frequency was hourly.

Figure 4 presents the FC3D model performance. The data and metrics used to obtain the new reconstruction discrepancy were explained in the paragraph above. It can be noticed that inspections (thicker green lines) did not present any change in the gas turbine’s operation, while major maintenance (the widest green line) revealed a considerably high deviation. According to these preliminary results, the inspections were no longer used in this study, as their contribution was not relevant. Next, several experiments were performed using this information and the models mentioned above, but only major maintenance was considered to assess the gas turbine behaviour.

Figure 5 summarises the effect of the replacements R during major maintenance in RD for the four tested autoencoder structures. R is displayed as a dashed vertical green line, while the RD values are displayed in blue. Furthermore, the centroid of each distribution, i.e., before and after maintenance, is displayed as a red cross. According to the obtained results, the fully connected autoencoder models showed a clearer distinction before and after maintenance operation than sparse models. Moreover, the best results were obtained using the FC3D model, in which a clear shift can be observed when a replacement procedure was carried out.

## 4. Discussion

The main goal of this study was to analyse the operational mode of a gas turbine working at full-load conditions after major maintenance using several autoencoder architectures. This analysis was carried out by combining two model structures: the two kinds of AE and their different internal dimensions. The input data did not present high dimensionality, as one of the aspects of using an AE architecture is that 5D representation cannot be properly shown. In this case, the data are transformed from 5D to either 2D or 3D, which makes their representation easier. This was not applied in this specific study; however, it will be considered for future research. Moreover, as explained in the introduction, an extensive number of studies exist in the literature tackling malfunctioning events in gas turbine systems based on autoencoder architecture models. Thereby, the application of these models was found to be useful to address gas turbine system management. To verify whether these kinds of models are able to extract the hidden patterns in the data and to determine which model performed the best for the proposed gas turbine, several kinds of AE with different latent dimensions were used.

Regarding the period of time taken to train the models, in the first-year data, it was revealed that the fully connected architecture was more effective for this specific gas turbine than sparse since the RD presented lower values (see Figure 5). Moreover, the one presenting the best performance was the FC3D model since the training and validation data had the lowest dispersion.

Although feature extraction did not benefit sparse architecture, a clear result was observed, which was the deviation of the operation point after maintenance performance, even though performance could be maintained. In terms of the RD, a positive slope was observed after the maintenance operation regarding the data distribution centroid for all four tested models. The most evident distinction was found in the FC models, or more precisely in the FC3D model, where a clear shift was present, and a more subtle one was also shown in sparse models. The detection of this deviation is crucial to understand how maintenance affects gas turbine performance and when an update of the internal operating models is needed.

The results presented in this study are consistent with previous outcomes in the same gas turbines. In [21], we presented a soft sensor that is able to detect drift in the compressor data. This research went a step further by determining the operational change after major maintenance.

## 5. Conclusions

This study aimed to analyse the condition and performance of a gas turbine after a maintenance operation using two different autoencoder architectures. The main goals were to assess the performance of the AE models in order to evaluate their feature extraction capabilities and to determine deviations in the system behaviour based on these models. As shown in the results, and further explained in the discussion section, the analysis revealed a clear slope regarding the gas turbine’s operation after a replacement (R) driven by proper feature extraction from AE-architecture-based models. Thereby, AE is able to learn complex features of data and capture significant operational changes in industrial gas turbines.

An interesting consideration of this analysis is that the data used were obtained from real plants operating at different modes, which led us to carry out an exhaustive analysis of the data to determine the most suitable filters and points to properly train every model.

In addition, all the data used to perform the study were only provided by the compressor, which is one of the main components of gas turbines. However, these results would allow us to extrapolate the analysis to the whole system. A further extension of this study could be to include some turbine data to assess more maintenance and malfunctioning events.

For further research, two clear lines can be followed. On the one hand, the analysis of latent space can be performed. In this study, the main analysis involved the performance of the AE itself. However, the autoencoder is able to encode the 5D input data to either 3D or 2D data. By definition, the autoencoder is able to maintain the relationship between the input variables and make it easier to visualise the data. This would lead to new and significant results.

On the other hand, a prognostic indicator of how the system will modify its operation according to its current behaviour can be obtained based on this study. As mentioned before, further information about the maintenance, faults, and unexpected events, combined with the combustion chamber and turbine information, can lead to more insights into component knowledge. Thereby, other components of gas turbines can be assessed, and the analysis can be performed on the whole system using the methodology of this study.

## Figures and Tables

**Figure 1 sensors-23-01236-f001:**
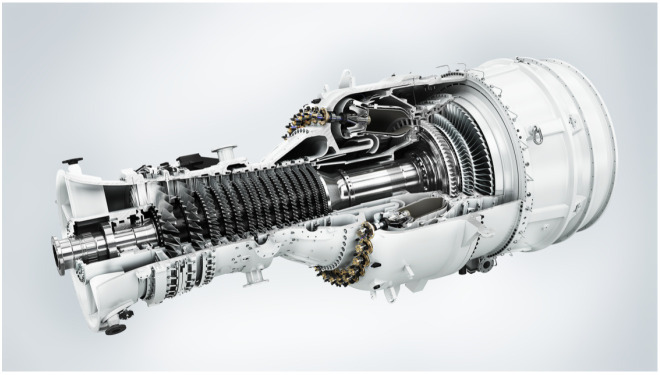
A unit of the gas turbine model used to perform the current analysis.

**Figure 2 sensors-23-01236-f002:**
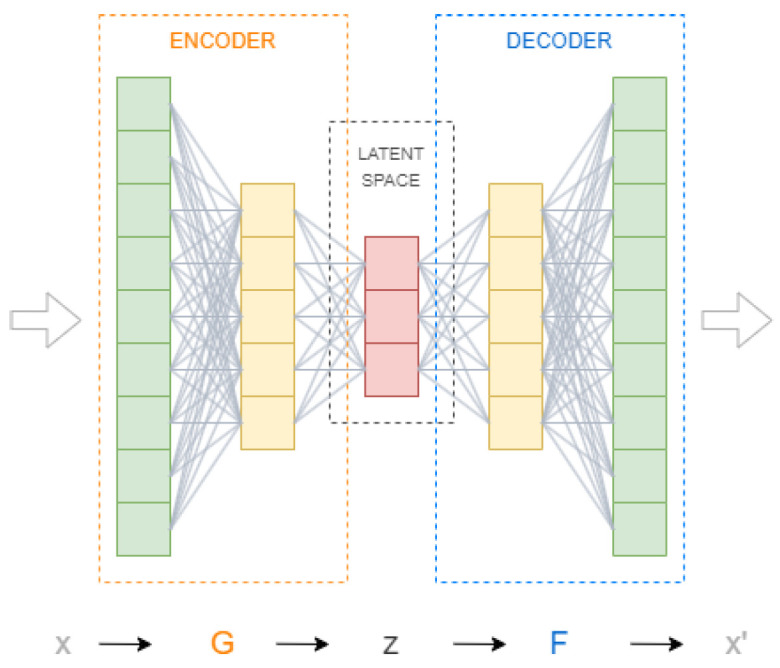
General example of an autoencoder with a three-dimensional latent space. Network G is an encoding function z=G(x), where *x* is the model input, and *z* are named latent variables. The decoder *F* is defined to reconstruct the encoded signal, $x^{\prime }=F\left(z\right)$. The set of weights for both networks is simultaneously learned by minimising a loss function $\epsilon =d(x,x^{\prime })$ according to some distance metric.

**Figure 3 sensors-23-01236-f003:**
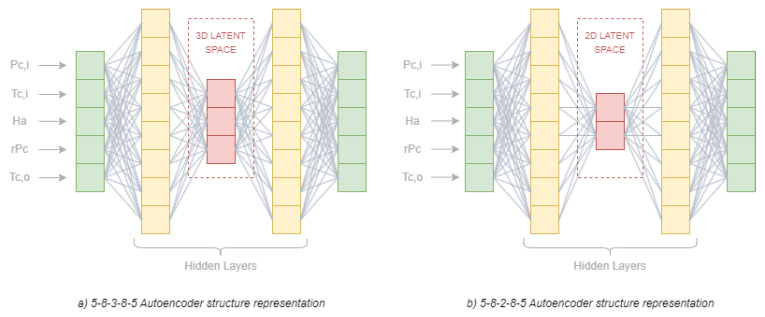
Representation of the two autoencoder structures employed to model a fresh gas turbine’s compressor.

**Figure 4 sensors-23-01236-f004:**
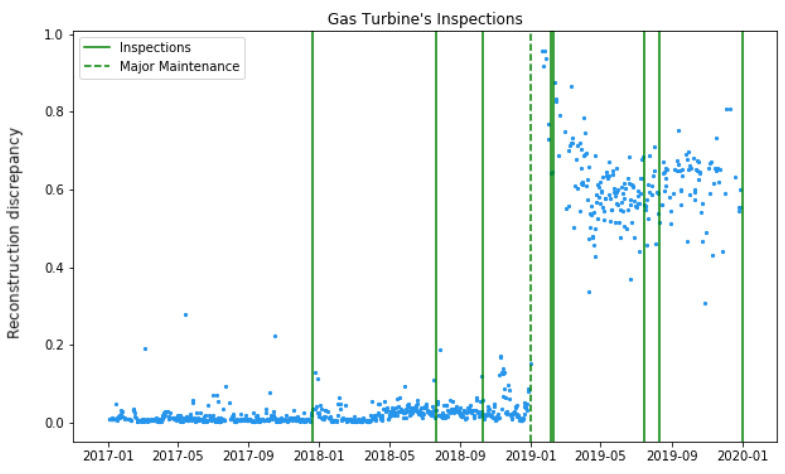
Maintenance dataset together with Fresh Reconstruction Discrepancy of FC3D model.

**Figure 5 sensors-23-01236-f005:**
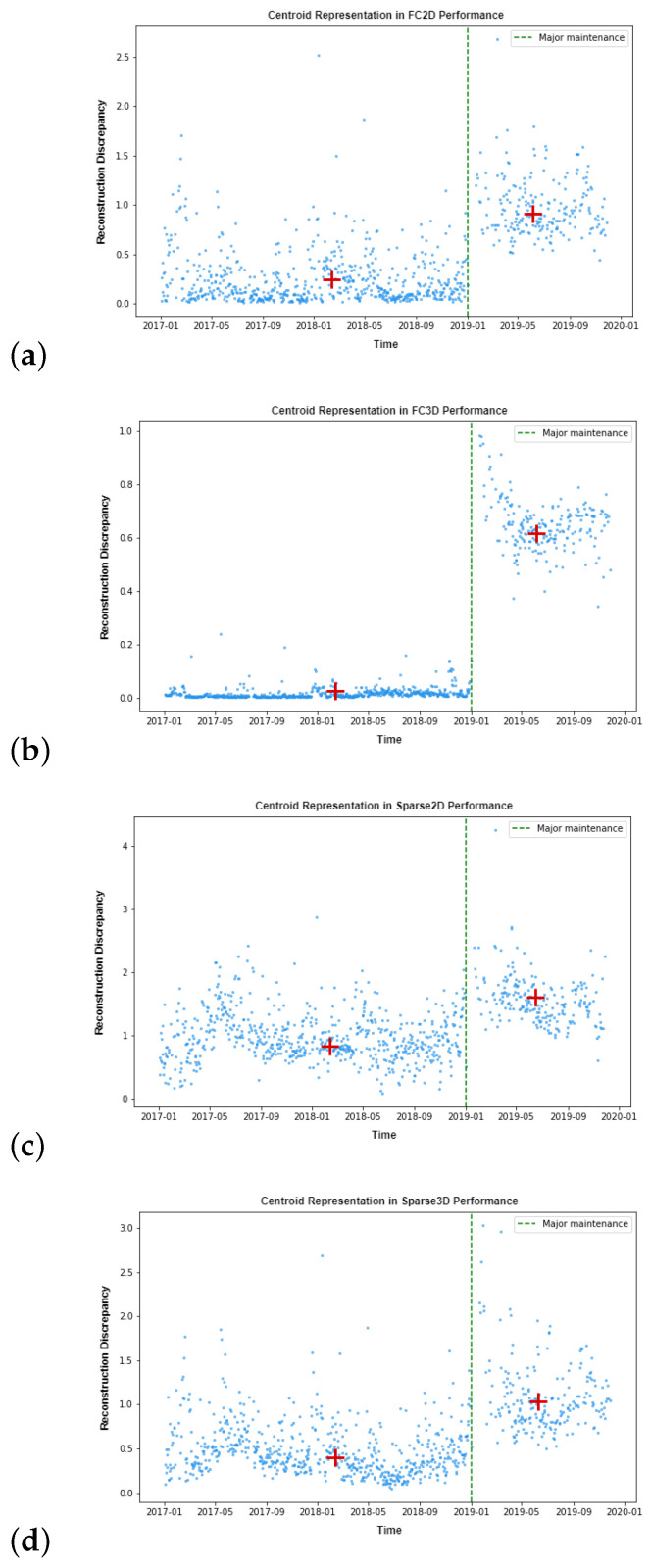
Major maintenance together with new reconstruction discrepancy in combination with the two AE structures and the two latent space dimensions: (**a**) FC2D model performance; (**b**) FC3D model performance; (**c**) Sparse2D model performance; (**d**) Sparse3D model performance.

**Table 1 sensors-23-01236-t001:** The combinations of the two considered AE types and the two latent dimensions.

		Latent Dimension	
		*2D*	*3D*
**AE structure**	Fully Connected	FC2D	FC3D
	Sparse	Sparse2D	Sparse3D

## Data Availability

Restrictions apply to the availability of these data. Data were obtained from Siemens Energy and are available from the authors with the permission of Siemens Energy.

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
