# Peer review of "Analysis of Gas Turbine Compressor Performance after a Major Maintenance Operation Using an Autoencoder Architecture"

_sensors, 2023, doi:10.3390/s23031236_

Round 1

Reviewer 1 Report

The paper presents a method to analysis the performance of a gas turbines after a maintenance operation using two different autoencoder architectures. On the whole, this paper lacks innovation. There is nothing new in the analysis method. It only shows the analysis of gas turbine data with conventional AE. In general, AE commonly used to analyze high-dimensional complex parameters, but only five parameters are used in in this analysis model. Unfortunatelythe article did not explain on it. In the analysis case, the super parameters of the used AE model and the comparison with other models are not shown.

Reviewer 2 Report

Comments to the authors

The article entitled “Analysing Gas Turbine Performance After a Major Maintenance Operation Using an Autoencoder Architecture” is a significant scholarly piece in the field of architectural industry. The authors have tried to prepare the manuscript well. However, this paper has colossal flaws, which are required to be addressed before publishing in this journal. I suggest for a ‘major revision’ and hence, add my comments as follows:

-In the first sentence of the abstract, using the word ‘way decisions’ is not correct. Please correct it and the sentence as well.

-please provide policy implications in the last part of the abstract.

-authors have not highlighted their study’s contribution to the existing pieces of literature. Please add specific contributions.

-please add literature review section and find out the research gap.

-discussion part is very poor. Please the add relevant issues and represent previous studies, which are coherent and incoherent with your study’s findings and why?

-conclusion section is also poor. Please rewrite it highlighting the brief results, significance of the study and people’s welfare relating to it.

-finally, authors should check the English language properly in using academic words.

Good luck!

Reviewer 3 Report

1) In Figure 2, there is no specific data in Figure a and Figure b, which should be lost

2) In line 171, it is determined experimentally by comparing the sensor diagram of gas turbine, and set it to be 7 times higher or lower than the median value corresponding to the upper or lower boundary. Why is it seven times? Any specific reasons?

3) Figure 3 shows the training set, testing set and post maintenance set. What is the logical relationship between the three? Where do the three data come from? How to obtain these data?

4) What is the difference between Figure a, Figure b, Figure c and Figure d for the data in Figure 5? How do they get their data?

5) There is no original data of the equipment in the article. Is there a more detailed data source process and analysis process? It is better to put some words in the appendix of the article.

Reviewer 4 Report

This research is study on the "Analysing Gas Turbine Performance After a Major Maintenance Operation Using an Autoencoder Architecture".

1- The title is not clear. Please check the  grammar and modify it.

2- Please add the image of gas turbine. 

3- More citations related to the analysis of gas turbine performance should be added.

4- Improve the quality of image in Fig. 1, 2, 3, 4,.. Make them more easy for understanding.

5- The discussion section is too short. It's not supported to this research. It should be written again.

Round 2

Reviewer 1 Report

The research topic of the article is the most concerned problem in the field of equipment management. Although the innovation of the method proposed in the article is not satisfactory, it provides an exploration way for this hot engineering practical problem. The revised version of the article has a clearer description of the contributions made in this article and has answered the questions in a way. Before accepting, the original intention of selecting AE model in the case of low-dimensional data needs further explanation. Moreover, there is still a lack of results comparison of relevant methods.

Reviewer 2 Report

Dear authors,

Greetings! Thanks for addressing the issues raised by me.

Thanks!

Reviewer 3 Report

1) In Figure 2, there is no specific data in Figure a and Figure b, which should be lost. The author only answers the questions without modifying the content.

2)In Page 7,“this can be noticed next in Figure [? ]”,the Figure number is missing.

3) How is the data in Figure 4 obtained? Why did the data change significantly in the first half of 2019?

4)The question mentioned in the last round "The original data of the equipment can't be seen in the article. Is there a more detailed data source process and analysis process? If there is any, it is better to put it in the appendix of the article". The author said that the data has privacy, but without specific data, these analysis will become meaningless, such as water without a source, trees without roots.

Round 3

Reviewer 3 Report

Many of the questions mentioned in the last round of review have been answered, and there are still questions to be improved in the paper, as follows:

(1)The various parameter data in the encoder and decoder modules in Figure 2 need to be filled in.

(2)Various parameter data of sensors input, 3D LATENT SPACE, 2D LATENT SPACE and output module in Figure 3 need to be filled in.
